# Effects of (a Combination of) the Beta_2_-Adrenoceptor Agonist Indacaterol and the Muscarinic Receptor Antagonist Glycopyrrolate on Intrapulmonary Airway Constriction

**DOI:** 10.3390/cells10051237

**Published:** 2021-05-18

**Authors:** Harm Maarsingh, Anouk Oldenburger, Bing Han, Annet B. Zuidhof, Carolina R. S. Elzinga, Wim Timens, Herman Meurs, Ramadan B. Sopi, Martina Schmidt

**Affiliations:** 1Department of Pharmaceutical Sciences, Lloyd L. Gregory School of Pharmacy, Palm Beach Atlantic University, West Palm Beach, FL 33416, USA; 2Department of Molecular Pharmacology, University of Groningen, 9713 AV Groningen, The Netherlands; oldenburgeranouk@gmail.com (A.O.); bigwhite.han@gmail.com (B.H.); a.b.zuidhof@rug.nl (A.B.Z.); c.r.s.elzinga@rug.nl (C.R.S.E.); h.meurs@rug.nl (H.M.); ramadan.sopi@uni-pr.edu (R.B.S.); 3Groningen Research Institute of Pharmacy, University of Groningen, 9713 AW Groningen, The Netherlands; 4Groningen Research Institute of Asthma and COPD, University of Groningen, University Medical Center Groningen, 9713 AV Groningen, The Netherlands; w.timens@umcg.nl; 5Department of Pathology and Medical Biology, University of Groningen, University Medical Center Groningen, 9713 GZ Groningen, The Netherlands; 6Department of Premedical Courses, Faculty of Medicine, University of Prishtina, 10000 Prishtina, Kosovo

**Keywords:** airway responsiveness, anticholinergic, β_2_-agonist, chronic obstructive pulmonary disease, glycopyrrolate, glycopyrronium, guinea pig, human, indacaterol, large airways, small airways

## Abstract

Expression of bronchodilatory β_2_-adrenoceptors and bronchoconstrictive muscarinic M_3_-receptors alter with airway size. In COPD, (a combination of) β_2_-agonists and muscarinic M_3_-antagonists (anticholinergics) are used as bronchodilators. We studied whether differential receptor expression in large and small airways affects the response to β_2_-agonists and anticholinergics in COPD. Bronchoprotection by indacaterol (β_2_-agonist) and glycopyrrolate (anticholinergic) against methacholine- and EFS-induced constrictions of large and small airways was measured in guinea pig and human lung slices using video-assisted microscopy. In guinea pig lung slices, glycopyrrolate (1, 3 and 10 nM) concentration-dependently protected against methacholine- and EFS-induced constrictions, with no differences between large and small intrapulmonary airways. Indacaterol (0.01, 0.1, 1 and 10 μM) also provided concentration-dependent protection, which was greater in large airways against methacholine and in small airways against EFS. Indacaterol (10 μM) and glycopyrrolate (10 nM) normalized small airway hyperresponsiveness in COPD lung slices. Synergy of low indacaterol (10 nM) and glycopyrrolate (1 nM) concentrations was greater in LPS-challenged guinea pigs (COPD model) compared to saline-challenged controls. In conclusion, glycopyrrolate similarly protects large and small airways, whereas the protective effect of indacaterol in the small, but not the large, airways depends on the contractile stimulus used. Moreover, findings in a guinea pig model indicate that the synergistic bronchoprotective effect of indacaterol and glycopyrrolate is enhanced in COPD.

## 1. Introduction

Chronic obstructive pulmonary disease (COPD) is a chronic inflammatory lung disease characterized by chronic progressive airflow obstruction that is only partially reversible and is caused by an abnormal inflammatory response to external stimuli, like cigarette smoking. The underlying airway inflammation in COPD is associated with epithelial changes, mucus hypersecretion, airway smooth muscle contraction, peribronchial fibrosis and emphysema, all contributing to development of airway obstruction [1,2]. There is a direct link between structural changes and airway function, where parenchymal destruction and biomechanical changes in the airway wall contribute to airway hyperresponsiveness [3]. According to the World Health Organization [4], COPD was the third common cause of death worldwide in 2020 and represents a big economic burden. Currently, no preventive or curative pharmacological treatment for COPD exists, therefore treatment is aimed at slowing down disease progression, reducing the exacerbation risk and alleviating disease symptoms in order to improve the quality of life [5]. Since airway obstruction is the main cause of dyspnea, bronchodilators have a main role in the pharmacological treatment of COPD [6].

The two most commonly used classes of bronchodilators are β_2_-adrenoceptor agonists (β_2_-agonists) and muscarinic M_3_ acetylcholine receptor antagonists (anticholinergics) [6]. β_2_-Agonists activate Gs-protein coupled β_2_-adrenoceptors resulting in the production of the second messenger cAMP, which induces airway smooth muscle relaxation [7,8]. Anticholinergics induce bronchodilation by antagonizing Gq-protein coupled muscarinic M_3_-receptors on airway smooth muscle, thereby attenuating the contractile response towards the increased cholinergic tone observed in COPD [9,10]. The increased cholinergic tone may also reduce the effectiveness of β_2_-agonists by protein kinase C-mediated desensitization of the β_2_-adrenoceptors in airway smooth muscle caused by crosstalk between muscarinic M_3_ and β_2_-adrenoceptors [11,12]. 

Autoradiographic visualization revealed that the density of β_2_-adrenoceptors increases with decreasing airway size [13], whereas the muscarinic M_3_-receptor density is higher in larger airways [14]. Moreover, the density of β_2_-adrenoceptors in the subsegmental bronchi is twice as high as the density of the muscarinic M_3_ receptors [15]. This would imply that β_2_-agonists would be more effective in relaxing small airways, however, anti-cholinergics block the contractile effect of endogenous acetylcholine and the distributions of cholinergic nerves is lower in smaller airways as well [16]. Determining the bronchodilator responses of large vs small airways towards β_2_-agonists and anti-cholinergics is important to optimize therapy of obstructive airways diseases directed towards small airways. 

The aim of this study was to compare the bronchoprotective effect of the long-acting β_2_-agonist indacaterol and the long-acting muscarinic M_3_-antagonist glycopyrrolate (also known as glycopyrronium) on large and small guinea pig intrapulmonary airways as well as in small intrapulmonary airways of COPD patients. Moreover, the synergistic effect of the combination of low concentrations of indacaterol and glycopyrrolate on the contractile response of large and small intrapulmonary airways was tested in a guinea pig model of COPD using repeated intranasal lipopolysaccharide (LPS) instillation. The bronchoprotective effect of glycopyrrolate on airway constriction induced by the muscarinic receptor agonist methacholine or by electrical field stimulation (EFS)—which activates cholinergic as well as non-adrenergic, non-cholinergic nerves—is similar between large and small airways. For indacaterol, the protective effect is most pronounced against EFS-induced constrictions of the small airways. Airway hyperresponsiveness in small intrapulmonary airways in COPD patients is normalized by indacaterol as well as by glycopyrrolate. The bronchoprotective effect the combination of indacaterol and glycopyrrolate is more pronounced in LPS-challenged guinea pigs compared to saline-challenged controls and synergistically normalizes airway responses in both the large and small airways.

## 2. Materials and Methods

### 2.1. Animals

Outbred, male, specified pathogen-free Dunkin Hartley guinea pigs (Harlan, Heathfield, UK) weighing 350–500 g were used. The animals were housed in pairs under a 12-h light/dark cycle in a temperature- and humidity-controlled room with food and tap water ad libitum. All animal care and experimental procedures complied with the animal protection and welfare guidelines and were approved by the Institutional Animal Care and Use Committee of the University of Groningen, The Netherlands, and are reported in compliance with the ARRIVE guidelines [17].

### 2.2. Guinea Pig Lung Slices

Precision-cut lung slices were prepared as described previously [3,18,19]. A 3% agarose solution in ultrapure water was prepared by heating it a microwave until fully dissolved. A double concentrated lung slice buffer (3.6 mM CaCl_2_, 1.6 mM MgSO_4_, 10.8 mM KCl, 232.8 mM NaCl, 2.4 mM NaH_2_PO_4_, 33.4 mM glucose, 52.2 mM NaHCO_3_, 50.4 mM Hepes, all of analytical grade; pH = 7.2) was 1:1 mixed with the 3% agarose solution after bringing both at 37 °C. Animals were sacrificed using an overdose of pentobarbital (Euthasol 20%, Produlab Pharma, Raamsdonkveer, The Netherlands) followed by exsanguination via the aorta abdominalis. Lungs were filled through a tracheal cannula at constant pressure with a low melting-point agarose (Gerbu Biotechnik GmbH, Weiblingen, Germany) solution (1.5%) in lung slice buffer (1.8 mM CaCl_2_, 0.8 mM MgSO_4_, 5.4 mM KCl, 116.4 mM NaCl, 1.2 mM NaH_2_PO_4_, 16.7 mM glucose, 26.1 mM NaHCO_3_, 25.2 mM Hepes, pH = 7.2) at 37 °C with 1 μM isoproterenol (Sigma-Aldrich, St-Louis, MO, USA) added to prevent post-mortem airway constriction. After filling, the lungs were covered with ice for 30 min to solidify the agarose for slicing. Lungs were removed and cylindrical tissue cores (diameter 15 mm) were prepared followed by slicing the tissue in ice cold lung slice buffer with 1 μM isoproterenol using a tissue slicer (CompresstomeTM VF- 300 microtome, Precisionary Instruments, San Jose CA, USA). Lung slices were cut at a thickness of 500 μm and washed several times with slicing buffer to remove debris and washout the isoproterenol. Slices were incubated overnight in a 6-wells plate in sterile minimal essential medium composed of lung slice buffer supplemented with 0.5 mM sodium pyruvate, 1 mM glutamine, MEM-amino acids mixture (1:50), MEM-vitamins mixture (1:100) and penicillin-streptomycin (1:100), all from Gibco, Carlsbad, CA, USA; pH = 7.2, at 37 °C in a CO2- and humidity-controlled atmosphere.

### 2.3. Lipopolysaccharide Instillation

Previous studies have demonstrated that guinea pigs challenged with lipopolysaccharide (LPS) are a good model for COPD as it induces various patho(physio)logical changes closely mimicking COPD [3,20,21]. For the guinea pig model of COPD, animals were repeatedly challenged with either LPS (COPD group) or saline (control group). At the start of the protocol, guinea pigs were randomly selected to be challenged by intranasal instillation of lipopolysaccharide (LPS, Sigma-Aldrich, St. Louis, MO, USA, 1 mg/200 μL in sterile saline) or 200 μL sterile saline (control group) twice weekly for 12 consecutive weeks [3,20]. Conscious guinea pigs were held in an upright position while the LPS solution was slowly instilled into the nose and animals were kept in the upright position for an additional 2 min to allow sufficient spreading of the fluid throughout the airways. Animal welfare was monitored by weighing the animals prior to each intranasal instillation; no animals needed to be withdrawn from the protocol. 24 h after the last challenge, animals were euthanized and precision-cut lung slices were prepared as described above.

### 2.4. Human Lung Slices

Peripheral lung tissue from COPD GOLD 1 (*n* = 5, all ex-smokers, two females) and GOLD 2 (*n* = 2, one ex-smoker and one current smoker) patients and from non-COPD control subjects (*n* = 5) was obtained from subjects (*n* = 4, three ex-smokers, one current smoker, one female) undergoing surgery for lung cancer using tumor-free tissue far from the tumor site and one control that was obtained from a non-transplanted donor lung (smoking status unknown). Median age was 64 (42–71) years for the COPD groups and 66 (42–69) years for the control group. All tissue was collected according to the Research Code of the University Medical Center Groningen (https://www.umcg.nl/SiteCollectionDocuments/English/Researchcode/umcg-research-code-2018-en.pdf, accessed on 19 April 2021) and national ethical and professional guidelines (‘Code of conduct’, Dutch federation of biomedical scientific societies, http://www.coreon.org, accessed on 19 April 2021). Precision cut lung slices were prepared as previously described [3,22]. After placing the lung tissue on a metal plate on ice, 2% low-melting agarose in slicing buffer was slowly and evenly injected at several sites of the tissue and the tissue was covered with ice for 15 min. Cylindrical cores of 15 mm in diameter were prepared, cut with a tissue slicer into 500 μm thin slices, and processed as described above for guinea pig lung slices.

### 2.5. Airway Responsiveness Measurements

After washing the slices in medium, individual slices were mechanically maintained with a Teflon ring with an inner diameter of 7 mm and covered with 1 mL of minimal essential medium. Only slices with approximately circular airways (longest/shortest diameter < 2) and with ciliary beating as an indication of viability were used. In guinea pig slices methacholine-induced contraction was measured both in large and small airways (in this species defined by diameters larger and smaller than 150 µm, respectively [3]), whereas in human slices only small airways (<500 µm) were studied. 

Two different types of stimuli to induce bronchoconstriction: direct muscarinic M_3_-receptor stimulation by methacholine and EFS-induced activation of cholinergic and non-adrenergic, non-cholinergic nerves. Methacholine was used, because—in contrast to acetylcholine—it is not rapidly hydrolyzed by cholinesterase and is selective for muscarinic receptors, whereas acetylcholine also activates nicotinic receptors [23]. Moreover, methacholine is often used in clinical settings to determine airway responsiveness, including in COPD patients [24], and we previously demonstrated small airway hyperresponsiveness in (a guinea pig model of) COPD using methacholine [3]. It can be envisaged that (functional) antagonism offered by indacaterol and glycopyrrolate against methacholine may differ from that of endogenous bronchoconstrictors. In addition, a different drug dose may be needed to block bronchoconstriction induced by vagal stimulation compared to bronchoconstriction induced by (inhaled) methacholine or acetylcholine [25]. Therefore, EFS was used as a stimulus to determine the effect of indacaterol and glycopyrrolate on the airway constriction induced by endogenous neurotransmitters released by cholinergic (acetylcholine) as well as non-adrenergic, non-cholinergic (neurokinins) nerves. 

Lung slices were pretreated with various concentrations of indacaterol and/or glycopyrrolate (Novartis, Basel, Switzerland) for 30 min and airway responses to increasing concentrations of methacholine (ICN Biomedicals, Zoetermeer, the Netherlands; 10^−8^–3·10^−3^ M, using cumulative concentrations in half−log increments) or electrical field stimulation (EFS, 0.5–62.5 Hz in doubling steps) was assessed using video-assisted microscopy (Eclipse TS 100, Nikon Instruments Europe BV, Amstelveen, The Netherlands) as previously described [3]. Image acquisition software (NIS-Elements 4.0 AR, Nikon Instruments Europe BV, Amstelveen, The Netherlands) was used to quantify airway luminal area. Images of the airways were acquired every 2 s during the whole course of the experiment, starting 2 min before the addition of any agent to allow for baseline measurements of the airway caliber. For each concentration of frequency, the maximal airway constriction was expressed as percentage of the initial (baseline) airway luminal area and plotted against that concentration/frequency. The maximal constriction (E_max_, % airway closure) and methacholine concentration or EFS frequency inducing 50% of the maximal response (pEC_50_ and F_50%_, respectively) were determined for each concentration- and frequency-response curve. 

Data are expressed as mean ± standard error of the mean (SEM). Figures were constructed using SigmaPlot 12.3 (Systat Software Inc, San Jose, CA, USA). Statistical differences were determined using a Mann-Whitney Test (SPSS Statistics 27, IBM, Chicago, IL, USA) and differences in maximal constriction, pEC_50_ and F_50%_ were considered to be statistical significant when *p* < 0.05.

## 3. Results

### 3.1. Effect of Indacaterol and Glycopyrrolate on Methacholine-Induced Constrictions of Large and Small Airways in Guinea Pig Lung Slices

The effects of treatment with different concentrations of the β_2_-agonist indacaterol or the anticholinergic glycopyrrolate on airway constrictions induced by the muscarinic receptor agonist methacholine were studied in large (>150 μm diameter) and small (<150 μm diameter) intrapulmonary guinea pig airways. Methacholine induced concentration-dependent constrictions of the large and small intrapulmonary airways (Figure 1).

#### 3.1.1. Effect of Indacaterol

In large intrapulmonary airways in guinea pig lung slices, indacaterol induced a concentration-dependent right-ward shift of methacholine-induced airway constrictions and reduced the maximal airway constriction (Figure 1A; Table 1). These effects were most pronounced at the highest concentration of indacaterol used (10 μM). The maximal constriction (E_max_) of control airways was reduced from 99.7 ± 0.6% to 71.5 ± 11.1% closure (*p* < 0.05) by 10 μM indacaterol which represents a protective effect of indacaterol of 28%. Moreover, 10 μM indacaterol provided a 32-fold reduction in the sensitivity (pEC_50_-value 4.9 ± 0.9) of large intrapulmonary airways compared to control (pEC_50_-value 6.4 ± 0.1; *p* < 0.001; Table 1). Whereas 10 μM indacaterol also reduced the sensitivity of the small intrapulmonary airways towards methacholine by 6-fold (*p* < 0.05), it did not significantly reduce the E_max_ (Figure 1B; Table 1). Overall, the bronchoprotective effects of indacaterol on the sensitivity and the maximal constriction were more pronounced in the large intrapulmonary airways compared to the small intrapulmonary airways (Figure 1C). These findings demonstrate that indacaterol offers a stronger bronchoprotective effect in the large airways than in the small airways as it relates to airway constrictions induced by muscarinic M_3_ receptor stimulation.

#### 3.1.2. Effect of Glycopyrrolate 

In the intrapulmonary large airways in guinea pig lung slices, glycopyrrolate induced a dose-dependent right-ward shift of methacholine-induced constrictions that was already observed at the lowest concentration studied (1 nM) and also reduced the maximal response in a dose-dependent manner (Figure 1D; Table 1). The highest concentration of glycopyrrolate tested, 10 nM, decreased the sensitivity 150-fold (*p* < 0.001) and reduced the maximal constriction to 58.9 ± 12.2% closure (*p* < 0.01), a 40% reduction (Figure 1D; Table 1). 

In the small intrapulmonary airways, glycopyrrolate also dose-dependently induced a right-ward shift of methacholine-induced constrictions that was already observed at the lowest concentration studied (1 nM) as well as the maximal response in a dose-dependent manner (Figure 1E; Table 1). The highest concentration of glycopyrrolate tested, 10 nM, decreased the sensitivity 60-fold (*p* < 0.001) and reduced the maximal constriction to 58.2 ± 10.0% closure (*p* < 0.01), a 38% reduction (Figure 1E; Table 1). 

When comparing the effect of the highest concentration of glycopyrrolate (10 nM) on large and small airways, a similar reduction in the maximal response was observed (40% and 38%, respectively), with a 2.5-fold larger shift in sensitivity in the large airways compared to the small ones (Figure 1F; Table 1). These findings demonstrate that glycopyrrolate offers a similar bronchoprotective effect in the large airways and in the small airways as it relates to airway constrictions induced by muscarinic M_3_ receptor stimulation.

#### 3.1.3. Indacaterol vs. Glycopyrrolate

We also compared the highest concentration of indacaterol (10 μM) with the highest concentration of glycopyrrolate (10 nM) used in guinea pig lung slices as it relates to their effect on methacholine-induced constrictions of large and small airways. Compared to indacaterol, glycopyrrolate induced a larger right-ward shift of the methacholine concentration-response curve and reduced the maximal constriction to a larger degree, which reached statistical significance in the small airways only (*p* < 0.05 both, Table 1). These findings indicate the glycopyrrolate protects better against muscarinic M_3_ receptor-mediated constrictions than indacaterol, particularly in the small airways.

### 3.2. Effect of Indacaterol and Glycopyrrolate on EFS-Induced Constrictions of Large and Small Airways in Guinea Pig Lung Slices

The effectiveness of indacaterol and glycopyrrolate in counteracting airway constrictions induced by endogenous neurotransmitters, including acetylcholine, was evaluated in guinea pig lung slices using EFS, which induced frequency-dependent constrictions of both large and small intrapulmonary airways (Figure 2, Table 2). No significant differences in the response to EFS between large and small airways were observed at any frequency in untreated guinea pig lung slices. 

#### 3.2.1. Effect of Indacaterol

Indacaterol dose-dependently reduced the sensitivity and the maximal response of both the large (Figure 2A) and small (Figure 2B) airways in guinea pig lung slices. In the large airways, the highest concentration of indacaterol (10 μM) reduced the maximal airway constriction from 75.1 ± 6.3% to 47.1 ± 4.4% (*p* < 0.05), whereas a higher frequency was needed to induce 50% of the maximal effect (F_50%_-value, *p* < 0.01, Figure 2A, Table 2). The 10 μM indacaterol also reduced the maximal constriction of the small airways (from 86.3 ± 5.4% to 20.8 ± 7.6%, *p* < 0.05) and increased the frequency needed to induce 50% of the maximal effect (*p* < 0.05, Figure 2B, Table 2). Interestingly, the protective effect of 10 μM indacaterol on the maximal constriction (E_max_) was more pronounced in the small airways compared to the large airways (*p* < 0.05, Figure 2C, Table 2). Thus, the indacaterol-sensitive portion of the constriction induced by the highest frequency (62.5 Hz) was 37% for the large airways and 76% for the small airways, whereas no difference in F_50%_-value was observed. Thus, indacaterol had a larger protective effect against airway constrictions induced by EFS than those induced by methacholine, particularly in the small airways.

#### 3.2.2. Effect of Glycopyrrolate

Glycopyrrolate dose-dependently reduced the sensitivity (F_50%_-value) and the maximal response (E_max_) of large airways (Figure 2D, Table 2) in guinea pig lung slices. In the presence of the highest concentration glycopyrrolate tested (10 nM), the maximal constriction reduced from 75.1 ± 6.3% to 23.3 ± 3.6% closure (*p* < 0.05) and a higher frequency was needed to induce 50% of the effect compared to control (24.3 ± 2.0 Hz and 13.5 ± 2.2 Hz, respectively, *p* < 0.05). Although the lower concentrations of glycopyrrolate (1 nM and 3 nM) hardly reduced EFS-induced constrictions in large airways, they offered a pronounced reduction in the small airways (Figure 2E, Table 2). 10 nM glycopyrrolate reduced the maximal EFS-induced constriction of the small airways from 86.3 ± 5.4% to 38.4 ± 11.5% closure (*p* < 0.05), but did not change the F_50%_-value (Table 2). The component of the EFS-induced constrictions sensitive to 10 nM glycopyrrolate was 69% in the large airways and 56% in the small airways (Figure 2F). Although the maximal bronchoprotective effect of glycopyrrolate is similar in large and small airways, the bronchoprotective effect of the anticholinergic occurs already at lower concentrations in the small airways.

#### 3.2.3. Indacaterol vs. Glycopyrrolate

In the large airways, the protective effect of 10 nM glycopyrrolate on the maximal EFS-induced constriction was more pronounced than that of 10 μM indacaterol (*p* < 0.05, Table 2) in guinea pig lung slices. No significant differences between the bronchoprotective effects of the highest concentrations of indacaterol and glycopyrrolate on EFS-induced constrictions were observed in the small airways (Table 2). These findings indicate that the anticholinergic glycopyrrolate offers better bronchoprotection against EFS-induced constriction of the large intrapulmonary airways than the β_2_-agonist indacaterol.

### 3.3. Effect of Indacaterol and Glycopyrrolate on Methacholine-Induced Constrictions Lung Slices Obtained from COPD Patients

Maximal methacholine-induced airway constriction of small airways (<500 μm) in lung slices obtained from subjects with GOLD I and II COPD was 1.5-fold increased as compared to non-COPD controls (67.9 ± 3.2% and 45.8 ± 11.1% closure, respectively, *p* < 0.05), without a significant difference in the pEC_50_ values (Figure 3, Table 3). Treatment with 10 μM indacaterol prevented the airway hyperresponsiveness in COPD patients to only 32.6 ± 13.0% closure (*p* < 0.01). Indacaterol also caused a 22-fold decrease in the sensitivity (pEC_50_-value) towards methacholine compared to COPD controls (*p* < 0.01, Table 3). Treatment with 10 nM glycopyrrolate similarly counteracted methacholine-induced airway hyperresponsiveness in COPD: a reduction of the airway hyperresponsiveness to the level of non-COPD controls (41.1 ± 11.2% closure; *p* < 0.01) and a 51-fold decrease in the sensitivity towards methacholine (*p* < 0.001, Figure 3, Table 3). Indacaterol as well as glycopyrrolate even reduced the sensitivity of the COPD airways by 18-fold and 42-fold compared to non-COPD controls (*p* < 0.05 and *p* < 0.01, respectively, Table 3). These findings demonstrate that muscarinic M_3_ receptor blockade as well as β_2_-adrenoceptor stimulation is effective in reversing airway hyperresponsiveness of small intrapulmonary airways in COPD patients.

### 3.4. Effect of (the Combination of) Low Concentrations of Indacaterol and Glycopyrrolate on Methacholine-Induced Airway Constrictions in a Guinea Pig Model of COPD

We studied the effect of low concentrations of indacaterol (10 nM) and glycopyrrolate (1 nM) alone and in combination on methacholine-induced constrictions of large airways as well as small airways in lung slices obtained from guinea pigs that were challenged twice weekly with either saline (control) or LPS (COPD model). 

#### 3.4.1. Effect in Large and Small Intrapulmonary Airways of Saline-Challenged Animals

In lung slices obtained from saline-challenged guinea pigs, the maximal constriction of the large airways was 1.9-fold higher compared to the small airways (*p* < 0.001) with a 4-fold higher sensitivity towards methacholine (*p* < 0.05, Figure 4, Table 4). The low concentrations of indacaterol (10 nM) or glycopyrrolate (1 nM) alone did not significantly change the airway responsiveness or sensitivity towards methacholine of the large (Figure 4A) or small (Figure 4B) airways of saline-challenged animals (Table 4). However, the combination of indacaterol and glycopyrrolate did reduce the airway sensitivity of the large airways by 5-fold (*p* < 0.01, Figure 4A, Table 4). The combination did not significantly affect methacholine-induced responses of the small intrapulmonary airways of saline-challenged guinea pigs (Figure 4B, Table 4). 

#### 3.4.2. Effect in Large and Small Intrapulmonary Airways of LPS-Challenged Animals

The maximal constriction of large intrapulmonary airways of LPS-challenged guinea pigs was 1.4-fold higher compared to the small airways (*p* < 0.001) with a 4-fold higher sensitivity towards methacholine (*p* < 0.05, Table 4). In contrast to the saline-challenged animals, indacaterol alone as well as glycopyrrolate alone reduced the sensitivity towards methacholine of the large airways by 4-fold (*p* < 0.05 each), whereas the combination caused a 16-fold reduction (*p* < 0.001, Figure 4C, Table 4). 

Although indacaterol alone or glycopyrrolate alone did not affect maximal methacholine-induced airway closure or the sensitivity towards methacholine in small airways of LPS-challenged guinea pigs, the combination reduced the maximal constriction from 69.9 ± 5.1% to 40.6 ± 7.1% (*p* < 0.05), a 42% reduction in airway responsiveness, as well as the sensitivity by 3-fold (*p* = 0.06, Figure 4D, Table 4). Interestingly, the contractile responses of the large and small airways in the LPS-challenged animals were fully normalized by the combined treatment of the low concentrations of indacaterol and glycopyrrolate and were almost identical to the airway responses of the saline-challenged animals in the presence of the combined treatment (Figure 4E,F, Table 4). Taken together, the airways of the LPS-challenged guinea pigs are more sensitive to the combined treatment with low concentrations of indacaterol and glycopyrrolate than the airways of saline-challenged animals. 

## 4. Discussion

Inhaled (long-acting) muscarinic M_3_-receptor antagonists and (long-acting) β_2_-agonists are the most commonly used bronchodilators for the treatment of COPD [6]. Where β_2_-agonists act as functional antagonist irrespective of the contractile stimulus, the anticholinergics particularly counteract the bronchoconstriction induced by the cholinergic tone, which is elevated in COPD [10,16]. Since the density of the β_2_-adrenoceptors and the muscarinic M_3_-receptor increases and decreases, respectively, as the airway size decreases [13,14], the bronchoprotection offered by β_2_-agonists and anticholinergics could potentially differ for large compared to small intrapulmonary airways which could impact the pharmacological treatment of respiratory diseases. 

We found that the degree of bronchoprotection offered by the anticholinergic glycopyrrolate against the muscarinic M_3_-receptor agonist methacholine was similar for large airways and small airways in guinea pig lung slices. Thus, glycopyrrolate induced a similar right-shift in the concentration-response curve of methacholine and a similar reduction in the maximal contractile response. There is a considerable receptor reserve regarding the muscarinic M_3_-receptors on airway smooth muscle in large airways for acetylcholine [26] and the full agonist methacholine [27]. Based on the fact that there is no real change in the effect of glycopyrrolate in the small airways compared to the large airways for methacholine, this would indicate that a receptor reserve is also present in the small airways. Glycopyrrolate was also able to greatly reduce the airway constrictions induced by EFS, which promotes the release of endogenous acetylcholine from nerve ending, but also of contractile mediators from excitatory non-adrenergic, non-cholinergic nerves [28]. The glycopyrrolate sensitive component of the maximal constriction induced by EFS stimulation was 69% in the large airways and 56% in the small airways, indicating that acetylcholine is the main endogenous contractile agent following EFS in guinea pigs. Although there was no difference in the bronchoprotective effect of the highest glycopyrrolate concentration used (10 nM) between large and small airways, the small were more sensitive to lower concentrations of glycopyrrolate (1 and 3 nM) than the large airways. These findings could be explained by a difference in receptor reserve for acetylcholine due to the lower density of M_3_-receptors in the small airways. However, since there was no significant difference in responsiveness to low concentrations of glycopyrrolate between large and small airways in the constrictions induced by methacholine, the difference in sensitivity towards low concentration of glycopyrrolate could also be explained by potential differences in EFS-induced release of contractile mediators in the large airways compared to the small airways. Taken together, despite the lower density of muscarinic M_3_-receptors in the small airways [14], there is no difference in the maximal bronchoprotective effects of anticholinergics in small airways compared to large airways. This is important, since we have previously observed that there is small airway hyperresponsiveness to methacholine in lung slices of COPD patients as well as in lung slices obtained from a guinea pig model of COPD [3]. 

Bronchodilator responses to β_2_-agonists, including indacaterol, on pre-constricted small airways] and to the anticholinergic tiotropium on carbachol-induced constriction of small airways [29] has been demonstrated in human and rat lung slices, but no comparison with larger airways was made. Similarly, the response to the anticholinergic tiotropium was demonstrated in small airways only in our current study, the response to indacaterol did differ between large and small airways, but was also dependent on the stimulus used. Indacaterol had a stronger protective effects against methacholine-induced constriction of the large airways compared to small airways, which seems at odds with the lower β_2_-adrenoceptor expression in large vs small airways [13]. However, indacaterol did protect more against EFS-induced constriction of small airways, in line with the higher receptor expression. Since indacaterol is a partial agonist with an intrinsic activity of 0.73 compared to the full agonist isoproterenol [30], changes in receptor expression in relation to airway smooth muscle mass as well as receptor desensitization are expected to affect its bronchodilatory response. The fact that the small airways are more sensitive to the bronchoprotective effect of lower concentrations of glycopyrrolate against EFS-induced constrictions than the large airways suggests less desensitization of the β_2_-adrenoceptor by acetylcholine-activated M_3_-receptors, which would explain the stronger bronchoprotective effect of indacaterol in the small airways compared to the large airways. However, the β_2_-adrenoceptor-mediated response in the small airways was not reduced compared to the large airways when using methacholine as a contractile agent instead of EFS. Therefore, the difference in responsiveness to indacaterol in the small airways between methacholine and EFS may result from the fact that the EFS-induced constrictions are caused by a combination of cholinergic and non-adrenergic, non-cholinergic receptor activation whereas methacholine only activates muscarinic receptors. This suggests that non-adrenergic, non-cholinergic mediators may preserve more β_2_-adrenoceptor function than methacholine.

Calzetta et al. compared the bronchodilator effects of olodaterol and tiotropium on isolated subsegmental bronchial rings and small intrapulmonary human airways in lung slices precontracted with a submaximal concentration of methacholine [31]. No significant differences were observed between the different airway sizes as it relates to the maximal bronchodilator response or the pEC_50_-value to either olodaterol or tiotropium [31]. However, since different experimental techniques were used to determine the airway contraction of the larger airways (airway tone of isolated bronchial rings) and the small airways (airway luminal area in intact lung tissue slices), the responsiveness (E_max_) of large and small airways cannot be compared. Since we used the same technique to study the constriction of large and small airways while normally embedded in lung tissue, the E_max_ and pEC_50_-values can be compared. Moreover, the fact that both airways sizes were still embedded in the lung slice ensured that morphological and functional characteristics were preserved at both the macroscopic and microscopic level. When comparing the highest concentration of indacaterol with the highest concentration of glycopyrrolate, no statistical significant differences were observed on the bronchoprotective effect against methacholine or EFS on either the large or the small airways. This is in line with the fact that no statistical significant effects were observed on the maximal bronchodilator response to either olodaterol or tiotropium in the subsegmental human bronchi or the human small airways [31].

We previously demonstrated small airway hyperresponsiveness in lung slices obtained from COPD patients (GOLD I and II) compared to non-COPD control [3]. We now demonstrate for the first time that the airway hyperresponsiveness of small intrapulmonary airways in COPD can be fully reversed by indacaterol as well as glycopyrrolate. Each agent also reduced the sensitivity (pEC_50_-value) towards methacholine compared to the response observed in non-COPD controls. The prevention of small airway hyperresponsiveness in COPD by indacaterol as well as glycopyrrolate is in line with clinical studies using inhaled treatments. Thus, glycopyrrolate monotherapy in COPD patients improved trough forced expiratory volume in one second (FEV_1_) after 12 and 24 weeks, improved dyspnea (transition dyspnea index focal scores) and reduces the need for rescue medication [32], whereas monotherapy with indacaterol lowers the exacerbation rate and improves lung function of COPD patients as evident by an improvement of trough FEV_1_ and dyspnea [33,34].

In a guinea pig model of COPD, we observed hyperresponsiveness of particularly the small airways that was similar to that observed in small airways in lung slices obtained from COPD patients [3]. Using the same LPS-induced guinea pig model of COPD, we investigated the synergistic effect of low concentrations of indacaterol and glycopyrrolate. In saline challenged animals, the low concentration of indacaterol and glycopyrrolate alone did not significantly affect airway responsiveness to methacholine, although these concentrations did slightly affect the pEC_50_ values in the naïve animals in the multiple dose study. The low doses of indacaterol and glycopyrrolate alone were effective in reducing the sensitivity, but not the maximal responsiveness, to methacholine in the large airways of the LPS-challenged guinea pigs. These results indicate that the sensitivity to β_2_-agonists and anticholinergics was increased in the LPS model compared to saline controls, which could have important beneficial therapeutic implications in COPD. Importantly, combined treatment with indacaterol and glycopyrrolate also had stronger protective effects on airway responses of LPS-challenged guinea pigs than in saline-challenged controls. Whereas the combination changed the sensitivity (pEC_50_), but not maximal responses, in the large airways of saline-challenged animals only, in LPS-challenged animals the sensitivity was reduced in both large and small airways—with the protection being larger in the large airways. Moreover, the combined treatment also reduced small airway hyperresponsiveness by 42%—compared to a non-significant reduction of 4.6% and 18.8% by indacaterol and glycopyrrolate alone. These findings highlight the importance of targeting the small airways and improving the drug delivery to the small airways, particularly since most end-points in studies in patients do not see a synergistic effect between indacaterol and glycopyrrolate compared to the individual effects, although the combination is superior to the individual treatments are often observed [35]. However, synergistic effects between β_2_-agonists and anticholinergics on bronchodilator responses have been observed in animals, as well as in human tissue and in patients (see reference [36] for a review). Thus combinations of tiotropium/olodaterol [31], glycopyrrolate/indacaterol [37], aclidinium/formoterol [38,39], and umeclidinium/vilanterol [40] synergistically improve airway relaxation and lung function in human tissue and in patients. However, comparing the effects of the combined treatment on large and small airway constrictions using the same method and in intact lung tissue has not been published before. Receptor crosstalk between the muscarinic M_3_-receptor and the β_2_-adrenoceptor leads to uncoupling of the β_2_-adrenoceptor [12,41], which may be exaggerated due to the increased cholinergic tone in COPD [10]. This could explain our finding that the combination therapy had a larger effect in the LPS-challenged animals compared to the saline-challenged controls. 

The concentrations of indacaterol (10 nM) and glycopyrrolate (1 nM) used in this study to determine possible synergistic effects are expected to be in the range of the local concentration of these drugs upon inhalation. The bioavailabilities of indacaterol and glycopyrrolate are 57% and 43%, respectively [42,43]. Maximal plasma concentrations after 14 days of treatment (C_max,ss_) depend on the amount inhaled but range between ~0.5 nM and ~2.5 nM for each drug (150 or 300 μg indacaterol and 50, 100 or 200 μg glycopyrrolate) [44,45,46,47]. Following inhalation, local concentrations in the lung are expected to be higher than these Cmax values. Thus, the concentrations used in this study are in the expected concentration range in the lung, which underscores the importance of targeting small airways in the treatment of COPD.

We observed some differences between the responses of the naïve animals in the multiple dose protocol and the saline-challenged control animals in the combination protocol. Whereas the maximal methacholine-induced airway constriction of the small airways was significantly less than that of the large airways in the saline challenged animals, in line with our previous finding [3], the airway responsiveness of the small airways to methacholine was not significantly different compared to the large airways in the naïve animals. No differences in the response to EFS were observed between large and small airways in guinea pig lung slices. In contrast, studies in rat lung slices demonstrated that the maximal airway responsiveness to EFS [48] is greater in large airways compared to small airways, indicating species-specific responsiveness to EFS. Maximal responses to methacholine are the same in large airways compared to small airways in rat lung slices, although the sensitivity to methacholine (pEC_50_) of small airways is greater than that of large airway in rat lung slices [49]. A lower sensitivity to methacholine of large airways compared to large airways is also reported for mice [50], but different techniques were used to study airway responsiveness. The lack of a lower response of the small airways in the naïve animals compared to the saline-challenged controls could result from the fact that the naïve animals were younger at the time of the experiment (because they did not enter a 12-week protocol) or by the fact that the lungs had not been exposed to saline twice weekly for 12 weeks.

## 5. Conclusions

In naïve guinea pigs, the anticholinergic glycopyrrolate similarly protects against muscarinic M_3_-mediated as well as EFS-induced airway constriction of large and small intrapulmonary airways that are embedded in their normal matrix in lung slices, demonstrating that the lower expression of muscarinic M_3_-receptors as the airways get smaller does not result in a lower therapeutic effect of anticholinergics. The bronchoprotective effect of the β_2_-agonist indacaterol is similar for large airways independent on the stimulus used, but the protective effect on small airways is larger against EFS than against methacholine. Indacaterol and glycopyrrolate each also normalize hyperresponsiveness of small airways in lung slices obtained from COPD patients. The synergistic effect of the combination of low concentrations of indacaterol and glycopyrrolate is more pronounced in large and small airways in lung slices obtained from a guinea pig model of COPD than in control animals. Taken together, these findings show that the bronchoprotective effect of glycopyrrolate does not de-pend on airway size, whereas the protective effect of indacaterol is different for large and small airways depending on the contractile stimulus used. Moreover, the findings in the guinea pig model indicate that the synergistic bronchoprotective effect of indacaterol and glycopyrrolate is enhanced in COPD.

## Figures and Tables

**Figure 1 cells-10-01237-f001:**
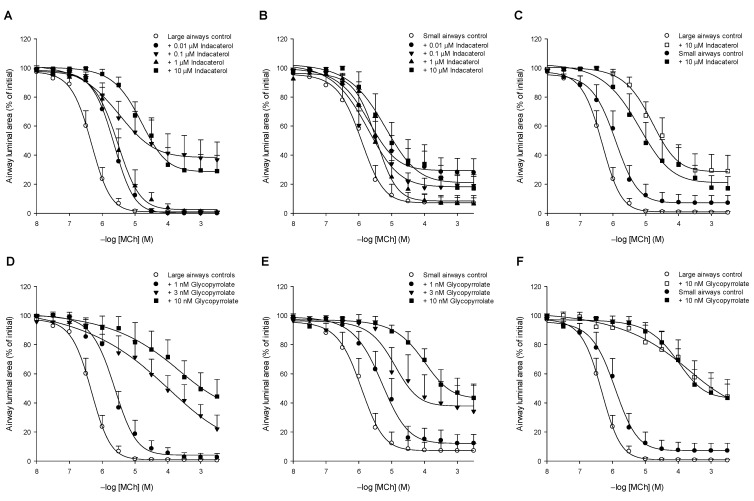
The effect of the β_2_-agonist indacaterol (**A**–**C**) and the anticholinergic glycopyrrolate (**D**–**F**) on methacholine-induced constrictions of large (**A**,**D**) and small (**B**,**E**) airways in guinea pig lung slices. Comparison of maximal drug effectiveness on large and small airways is shown in panels C and F. Lung slices were incubated with increasing concentrations of indacaterol (0.01, 0.1, 1 and 10 μM) or glycopyrrolate (1, 3 and 10 nM) and the constriction of large (>150 μm) and small (<150 μm) airways to increasing concentrations of methacholine (MCh) was determined by measuring the luminal area as a percentage of baseline. Data are represented as means ± SEM of 5–10 guinea pigs (see also Table 1).

**Figure 2 cells-10-01237-f002:**
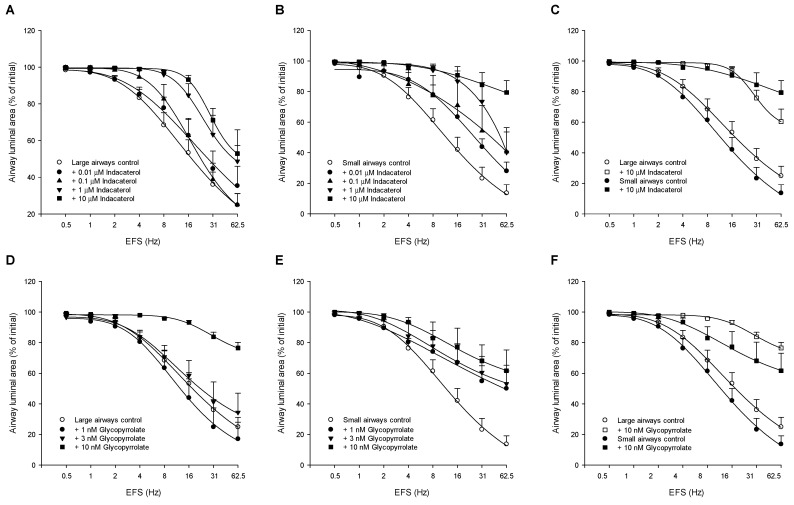
The effect of the β_2_-agonist indacaterol (**A**–**C**) and the anticholinergic glycopyrrolate (**D**–**F**) on EFS-induced constrictions of large (**A**,**D**) and small (**B**,**E**) airways in guinea pig lung slices. Comparison of maximal drug effectiveness on large and small airways is shown in panels C and F. Lung slices were incubated with increasing concentrations of indacaterol (0.01, 0.1, 1 and 10 μM) or glycopyrrolate (1, 3 and 10 nM) and the constriction of large (>150 μm) and small (<150 μm) airways to increasing frequencies (0.5–62.5 Hz) of EFS was determined by measuring the luminal area as a percentage of baseline. Data are represented as means ± SEM of 2–10 guinea pigs (see also Table 2).

**Figure 3 cells-10-01237-f003:**
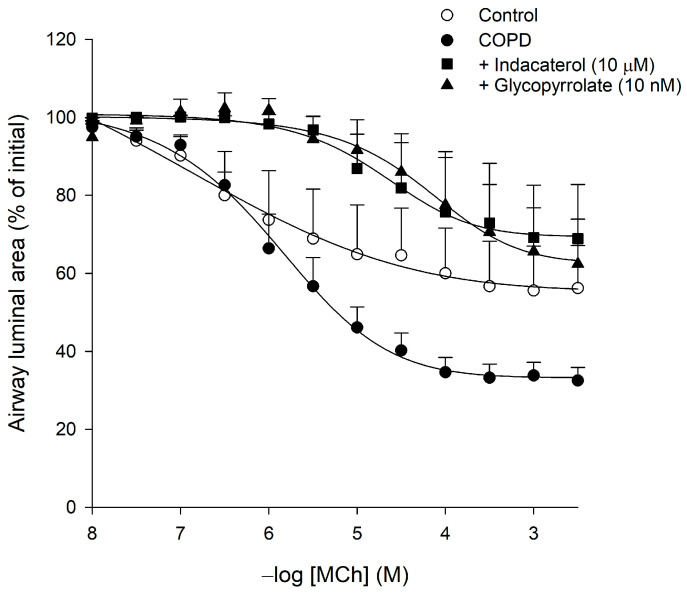
Methacholine (MCh)-induced constriction of small intrapulmonary airways (<500 μm) in human lung slices obtained from COPD GOLD I and II patients and non-COPD controls. The effects of the β_2_-agonist indacaterol (10 μM) and the anticholinergic glycopyrrolate (10 nM) on methacholine-induced airway constrictions ere determined in lung slices obtained from COPD patients. Airway constriction was determined by measuring the luminal area as a percentage of baseline. Data are represented as means ± SEM of 3–7 subjects (see also Table 3).

**Figure 4 cells-10-01237-f004:**
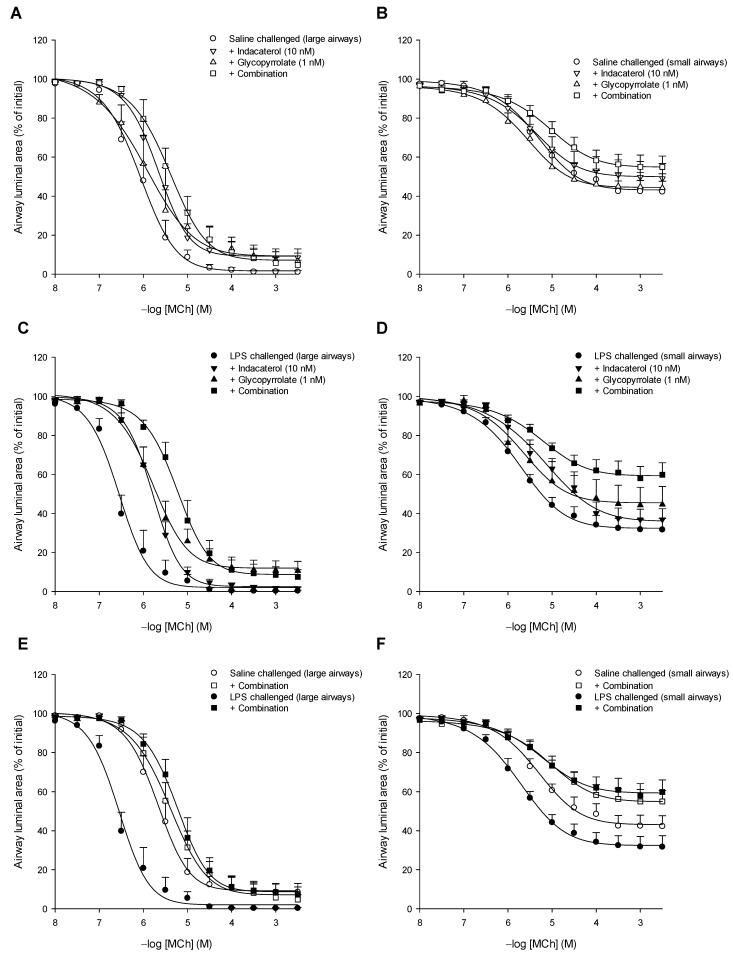
The effect of low concentrations of the β_2_-agonist indacaterol (10 nM) and the anticholinergic glycopyrrolate (1 nM) alone and combined on methacholine (MCh)-induced constrictions of large (**A**,**C**,**E**) and small (**B**,**D**,**F**) airways in lung slices obtained from guinea pigs that have been challenged twice weekly for 12 weeks with either saline (**A**,**B**) or LPS (**C**,**D**). Lung slices were incubated with indacaterol (10 nM), glycopyrrolate (1 nM) or the combination of both and the constriction of large (>150 μm) and small (<150 μm) airways to increasing concentrations of methacholine (MCh) was determined by measuring the luminal area as a percentage of baseline. Data are represented as means ± SEM of 6–10 guinea pigs (see also Table 4).

**Table 1 cells-10-01237-t001:** Effects of indacaterol and glycopyrrolate on methacholine-induced airway constriction of large and small airways in guinea pig lung slices.

	Large Airways			Small Airways		
Treatment	E_max_ (% Closure)	pEC_50_ (−log M)	*n*	E_max_ (% Closure)	pEC_50_ (−log M)	*n*
Control	99.7 ± 0.6	6.41 ± 0.14	10	93.5 ± 4.8	6.01 ± 0.19	10
0.01 μM indacaterol	101.3 ± 1.3	5.66 ± 0.17 **	5	73.1 ± 9.7	5.67 ± 0.19	8
0.1 μM indacaterol	73.3 ± 9.3 *	5.15 ± 0.40 **	6	83.0 ± 6.5	5.70 ± 0.20	8
1 μM indacaterol	99.3 ± 0.8	5.57 ± 0.17 **	8	94.1 ± 4.1	5.51 ± 0.21	8
10 μM indacaterol	71.5 ± 11.1 *	4.90 ± 0.19 ***	7	85.7 ± 8.5 ^†^	5.17 ± 0.30 *^,^^†^	7
1 nM glycopyrrolate	97.7 ± 2.9	5.61 ± 0.16 ***	7	89.2 ± 5.6	5.33 ± 0.21 *	8
3 nM glycopyrrolate	78.6 ± 8.3 *	4.68 ± 0.33 ***	8	66.8 ± 18.5 *	4.77 ± 0.28 **	6
10 nM glycopyrrolate	58.9 ± 12.2 **	4.21 ± 0.35 ***	7	58.2 ± 10.0 **	4.23 ± 0.24 ***	8

* *p* < 0.05, ** *p* < 0.01, *** *p* < 0.001 compared to control or the respective airway size, ^†^
*p* < 0.05 compared to 10 nM glycopyrrolate in the small airways. Data represents mean ± SEM of *n* experiments. Abbreviations: E_max_, maximal constriction induced by methacholine; pEC_50_, −log of the methacholine concentration inducing 50% of the maximal response.

**Table 2 cells-10-01237-t002:** Effects of indacaterol and glycopyrrolate on EFS-induced airway constriction of large and small airways in guinea pig lung slices.

	Large Airways			Small Airways		
Treatment	E_max_ (% Closure)	F_50%_ (Hz)	*n*	E_max_ (% Closure)	F_50%_ (Hz)	*n*
Control	75.1 ± 6.3	13.5 ± 2.2	10	86.3 ± 5.4	11.6 ± 2.3	9
0.01 μM indacaterol	64.6 ± 10.6	14.4 ± 3.7	4	71.9 ± 5.8	16.6 ± 3.8	5
0.1 μM indacaterol	74.9 ± 5.9	17.5 ± 2.7	4	59.3 ± 12.9 *	22.7 ± 7.6	5
1 μM indacaterol	51.3 ± 17.2	27.8 ± 4.6 *	4	40.3 ± 15.8 *	23.6 ± 5.8 *	5
10 μM indacaterol	47.1 ± 4.4 *	27.4 ± 1.5 **	4	20.8 ± 7.6 **^,†^	22.5 ± 4.8 *	4
1 nM glycopyrrolate	72.7 ± 10.9	12.8 ± 3.4	4	49.9 ± 11.4 *	10.9 ± 3.8	3
3 nM glycopyrrolate	53.6 ± 12.0	18.3 ± 7.9	4	46.9 ± 18.1	12.5 ± 4.8	2
10 nM glycopyrrolate	23.3 ± 3.6 **^,†^	24.3 ± 2.0 *	3	38.4 ± 11.5 *	15.0 ± 5.7	3

* *p* < 0.05, ** *p* < 0.01 compared to control or the respective airway size, ^†^
*p* < 0.05 compared to 10 μM indacaterol in the large airways. Data represents mean ± SEM of *n* experiments. Abbreviations: E_max_, maximal constriction induced by methacholine; F_50%_, EFS frequency inducing 50% of the maximal response.

**Table 3 cells-10-01237-t003:** Effects of indacaterol and glycopyrrolate on methacholine-induced airway constriction of human intrapulmonary airways obtained from non-COPD control subjects and COPD patients (GOLD I and II).

Treatment	E_max_ (% Closure)	pEC_50_ (−log M)	*n*
Non-COPD control	45.8 ± 11.1	5.68 ± 0.37	5
COPD (GOLD I and II)	67.9 ± 3.2 *	5.77 ± 0.18	7
+10 μM indacaterol	32.6 ± 13.0 ^‡^	4.43 ± 0.55 *^,‡^	3
+10 nM glycopyrrolate	41.1 ± 11.2 ^‡^	4.06 ± 2.0 **^,‡‡^	4

* *p* < 0.05, ** *p* < 0.01 compared to non-COPD control, ^‡^
*p* < 0.01, ^‡‡^
*p* < 0.001 compared COPD control. Data represents mean ± SEM of *n* experiments. Abbreviations: E_max_, maximal constriction induced by methacholine; pEC_50_, −log of the methacholine concentration inducing 50% of the maximal response.

**Table 4 cells-10-01237-t004:** Effects of indacaterol and glycopyrrolate on methacholine-induced airway constriction of large and small airways in guinea pig lung slices.

	Large Airways			Small Airways		
	E_max_ (% Closure)	pEC_50_ (−log M)	*n*	E_max_ (% closure)	pEC_50_ (−log M)	*n*
*Saline-challenged*						
Control	99.2 ± 0.9	6.13 ± 0.15	10	51.8 ± 7.8 ^###^	5.49 ± 0.16 ^##^	8
10 nM indacaterol	92.2 ± 4.5	5.76 ± 0.13	8	59.7 ± 5.0 ^###^	5.42 ± 0.10	6
1 nM glycopyrrolate	93.3 ± 4.1	6.06 ± 0.27	8	60.2 ± 7.3 ^###^	5.58 ± 0.13	6
Combination	95.7 ± 2.4	5.40 ± 0.14 **^,†^	9	46.7 ± 5.6 ^###^	5.16 ± 0.15	9
*LPS-challenged*						
Control	99.7 ± 0.2	6.46 ± 0.17	9	69.9 ± 5.1 ^###^	5.81 ± 0.10 ^#^	7
10 nM indacaterol	99.0 ± 0.9	5.80 ± 0.09 *	8	66.7 ± 6.4 ^###^	5.49 ± 0.22	8
1 nM glycopyrrolate	90.8 ± 5.2	5.82 ± 0.14 *	8	56.8 ± 9.3 ^##^	5.44 ± 0.21	8
Combination	92.9 ± 3.6	5.25 ± 0.12 **^,++,‡^	8	40.6 ± 7.1 *^,###,+^	5.34 ± 0.19	8

** p* < 0.05, ** *p* < 0.01 compared to control or the respective airway size and group (saline/LPS), ^#^
*p* < 0.05, ^##^
*p* < 0.01, ^###^
*p* < 0.001 compared to large airways of the same condition and group (saline/LPS), ^+^
*p* < 0.05, ^++^
*p* < 0.01 compared to 10 nM indacaterol of the same group (saline/LPS) and respective airway size, ^†^
*p* < 0.05, ^‡^
*p* < 0.01 compared to 1 nM glycopyrrolate of the same group (saline/LPS) and respective airway size. Data represents mean ± SEM of *n* experiments. Abbreviations: E_max_, maximal constriction induced by methacholine; pEC_50_, −log of the methacholine concentration causing 50% of the maximal response.

## Data Availability

The data used to support the findings of this study are available from the corresponding author.

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
