# Peer review of "Effects of (a Combination of) the Beta2-Adrenoceptor Agonist Indacaterol and the Muscarinic Receptor Antagonist Glycopyrrolate on Intrapulmonary Airway Constriction"

_cells, 2021, doi:10.3390/cells10051237_

Round 1

Reviewer 1 Report

The submitted manuscript entitled „ Effects of (a combination of) the beta2-adrenoceptor agonist in- dacaterol and the muscarinic receptor antagonist glycopyrrolate on intrapulmonary airway constriction” by Harm Maarsingh and colleagues describes the therapeutic combination of indacaterol and glycopyrrolate in different models in vivo and in vitro models. The manuscript improved during revision, contains important details, is well written, clear, and logical. However there are some important concerns.

Major points:

  1. Why did the authors use Metacholin to induce bronchoconstriction? Acethylcholin would fit better to the response initiated by the EFS.

Glycopyrrolate is a non-specific muscarinic antagonist. The two major muscarinic receptors in the lungs are M3 on the bronchial smooth muscle and the M2 on nerves. In small airways the inhibition by glycopyrrolate may result in a common concentration-response curve, whereas this seems not to be the case in large airways. This have to be analyzed and discussed in the manuscript. For small airways, with low or no neural influence I would expect a shift to the right in the concentration-response curve, due to the muscarinic antagonist glycopyrrolate.

The EFS of small airways seems to contract those airways much stronger than large airways. This is in contradiction to the cited reference 35. The author should discuss this issue.

In Figure 3 the authors measured the effect of indacaterol and glycopyrrolate alone and in combination on Mch induced bronchoconstriction in human lung slices. This was only performed in the slices obtained from COPD patients. Why are there no data for indacaterol and glycopyrrolate in human control lungs (non-COPD)? Otherwise, the authors cannot conclude, that indacaterol and glycopyrrolate can fully reverse AHR (line 304-305) as the corresponding data of control slices are missing.

The author should discuss the effect of relevant concentrations in the in vitro models, for example based on plasma concentrations found after oral application.

In figure 2 D-E the authors found small effects of glycopyrrolate (1nM and 3nM) in EFS, whereas the effect was stronger on MCh induced bronchoconstriction. The authors should discuss the role of the different neural mediators in more detail.

Minor points:

The authors should add a section about the statistical analysis. The authors mention the Mann-Whitney Test, as a non-parametric test. However, the authors did not mention if the data were corrected for multi-comparison. How did the authors calculate the four parameters of the sigmoidal curve? Did they use GraphPad Prism? This would allow to compare the curves directly. In some figures the curves should be compared for common max, min, and hill slope. Please comment!

Reviewer 2 Report

Maarsingh et al. Cells

This is an interesting study examining the effects of indacaterol, glycopyrrolate, or both on MAC- and EFS- stimulated small and large airway contraction in PCLS from GP or COPD/control patients. The question is important given each drug may regulate large and small airways differently, and with such knowledge a patient’s pulmonary function profile could enable a more informed management. Much new information is provided in this study, and the design could be replicated in future studies using PCLS from different animal models or human subjects to gain further insight into disease pathophysiology and drug effectiveness.

The study is rather robust, and although I would have preferred more data using human PCLS, the findings are more than sufficient to move the field.   I have no major concerns about the data or the conclusions, but do have a few comments regarding interpretation that, if addressed, could improve the paper.

  1. One premise I question is that variable receptor (beta-2-AR or m3mAChR) density likely influences the response of large vs small airways. This might be the case for the beta-2-AR, but it is likely that m3mAChR expression is never limiting, and that airways with lower m3 expression still have spare receptors.
  2. EFS in the slices seems to work well, something I would have worried about. However, IC50 values for indacaterol for most experiments seem high for a drug with a Ki of 5 nM; would this suggest some diffusion limitations in the slices? Do on/off rates for indacaterol confound the analysis?
  3. Given the protective effect of indacaterol on small airways is larger against EFS than against MCh, doesn’t this suggest a sufficient NANC component on smaller airways with EFS? Indacterol-mediated inhibition of MCh-mediated contraction would be limited by functional antagonism of the beta-2 by MCh. However, small airways subjected to EFS-stimulated (NANC-mediated contraction + ACh-mediated contraction) would not necessary be subject to significant functional antagonism.
  4. The cooperative effect of the 2 drugs at low concentrations is impressive and perhaps undersold here, as these data don’t really exist in in vivo human studies for numerous reasons, including dose-response data doesn’t really exist in human subjects. Indeed the cooperative effect of combined LABA+ICS or LABA+LAMA on FEV1 isn’t even significant; one needs to point to clinical outcomes to argue the effectiveness of combination drugs. And when one considerations the limitations of drug delivery to the small airways, in both COPD and asthma patients, the data here from lung slices may give some important insight into the function of these drugs in vivo.

Reviewer 3 Report

This manuscript describes the effects of two different bronchodilators (indacaterol, a beta2 agonist) and glycopyrrolate (a muscarinic receptor antagonist) on the constriction of airways (both human and from a guinea pig model) using lung slices and video-assisted microscopy.

The manuscript is scientifically sound, the conclusions justified by the results. 

I have a few issues I would like the Authors to address:

In clinical practice, indacaterol and glycopyrrolate are used at similar concentrations: The only available commercial preparation contains 84 µg indacaterol (m.w 392) and 43 µg glycopyrronium (m.w. 318) per dose. I am not clear why the concentrations used in this work differ by three orders of magnitude.

The statement that small airways in guinea pigs are defined as < 150 µm should be supported by a reference.

In figure C, panels C and F have different symbols for the same groups (although with different drugs); using the same 4 symbols in both panels would help clarity

I found paragraph 3.1.3 difficult to follow; would it be possible to partially rephrase it to make it clearer?

Figure 3 (at least in the file I downloaded ) is of poor graphic quality compared with the others.

Round 2

Reviewer 1 Report

The resubmitted manuscript has strongly improved during revision.

Minor comment:

The use of methacholine instead of acetylcholine should be explained with pro and cons. The answer so far was not sufficient. 

It’s a pity, that the number of control tissue slices was not sufficient to prove the effect of the therapeutic combination there.  

Author Response

Response to Reviewer 1:

The resubmitted manuscript has strongly improved during revision.

We thank the Reviewer for their constructive comments and feedback regarding this manuscript and revision.

Minor point:

Point 1: The use of methacholine instead of acetylcholine should be explained with pro and cons. The answer so far was not sufficient.

Response 1: We have compared and contrasted the use of methacholine and acetylcholine in the revised manuscript (page 4, line 167-181) and hope that we have now satisfactorily addressed the concern raised by the Reviewer. In short, we have added that the pros of methacholine include their higher resistance to metabolism by cholinesterases and their increased selectivity for muscarinic receptors compared to acetylcholine. We also included that a con is that findings on (functional) antagonism against methacholine may differ from that against acetylcholine. It has been postulated that the dose of a bronchodilator to block the bronchoconstriction induced by (inhaled) acetylcholine or methacholine may differ from that needed to block bronchoconstriction induced by vagal stimulaton (PMID 21198547).

Point 2: It’s a pity, that the number of control tissue slices was not sufficient to prove the effect of the therapeutic combination there. 

Response 2: We agree with the Reviewer.